# Prevalence of *Trichomonas vaginalis* Infection in Women Screened for Precursor Lesions of Cervical Cancer in a Brazilian Population

**DOI:** 10.3390/microorganisms12102032

**Published:** 2024-10-08

**Authors:** Marina de Paula Salomé dos Santos, Bruna Ribeiro de Andrade Ramos, Maria Luiza Cotrim Sartor de Oliveira, Andréa da Rocha Tristão, Márcia Guimarães da Silva

**Affiliations:** 1São Paulo State University, Unesp, Botucatu Medical School, Botucatu 18618-687, SP, Brazil; marina.salome@unesp.br (M.d.P.S.d.S.); bruna.ra.ramos@unesp.br (B.R.d.A.R.); maria.lcs.oliveira@unesp.br (M.L.C.S.d.O.); andrea.tristao@unesp.br (A.d.R.T.); 2Jaú Medical School, Western São Paulo University—UNOESTE, Jaú 17213-700, SP, Brazil

**Keywords:** sexually transmitted infection, *Trichomonas vaginalis*, trichomoniasis, pap smear, cervical cytology

## Abstract

*Trichomonas vaginalis* infection is one of the most prevalent curable STIs. Although treatments are available, *T. vaginalis* infections pose a significant challenge, especially in resource-limited regions, as the prevalence of this STI is often unknown. We aimed to determine the prevalence of *Trichomonas vaginalis* infection in women screened for cervical cancer precursor lesions in Botucatu in São Paulo, Brazil. We conducted a descriptive and retrospective study that included 23,735 women who attended the cervical cancer screening program at health units in 2019 and 2022. Clinical and sociodemographic data were collected from the cancer information system (SISCAN) and test requisition forms. Descriptive analysis was conducted, and comparisons were performed using the X2 Test and Student’s *t*-test (SigmaPlot version 13.0). The prevalence of *T. vaginalis* infection was 0.84% in 2019 and 0.57% in 2022. The mean age of patients with trichomoniasis was 42 (±11.2) years; 75% self-reported as white, 43% were married or in a stable relationship, and 40% had not completed primary education. Regarding the vaginal microbiota, only 15.3% of the cytology exams with infection by *T. vaginalis* showed a predominance of lactobacilli species, while inflammation was present in 82% of the smears. Cytological analysis revealed precursor lesions of cervical cancer in 0.05% of patients with trichomoniasis, including ASC, LSIL, and HSIL. The study showed a low prevalence of infection with *T. vaginalis* in low-risk women screened for precursor lesions of cervical cancer in Botucatu in São Paulo, Brazil.

## 1. Introduction

According to the World Health Organization, more than 374 million new cases of curable sexually transmitted infections (STIs) are diagnosed globally each year in individuals of reproductive age [1]. *Trichomonas vaginalis* infection is one of the most prevalent curable STIs. Despite available treatments, *T. vaginalis* infections pose a significant challenge, especially in resource-limited regions, as they are often oligosymptomatic and underdiagnosed due to the scarcity of diagnostic methods in these settings [2].

Population-based studies in the United States observed rates of 2.3% among adolescents [3] and 3.1% among patients aged 14 to 49 [4]. In Brazil, prevalence rates vary widely, ranging from as low as 0.2% to as high as 19% in different settings [5,6,7,8,9]. A study conducted in Pará reported a prevalence of 0.68% using the Papanicolaou test [10], while Lima et al. [6] found a 10.5% prevalence in Pernambuco, with 35% of cases in rural areas. A systematic review showed higher prevalence in the north and northeast regions, ranging from 12.7% to 19%. The most frequently employed diagnostic methods were the Papanicolaou test and direct examination of vaginal content (35.7%), followed by polymerase chain reaction (28.7%) and culturing (18.7%) [11].

The most reported signs and symptoms of this infection include dyspareunia and vaginal discharge, which may be diffuse, malodorous, yellowish, or greenish, and is sometimes accompanied by vulvar irritation. Clinical examination may reveal cervical punctate hemorrhage, referred to as “strawberry cervix” [12]. The infection increases susceptibility to pelvic inflammatory disease in women [13,14] and reduces sperm viability in men, affecting reproduction [15]. Trichomoniasis can also complicate pregnancy, contributing to low birth weight, premature rupture of membranes, and preterm birth [16].

There are several methods that can be used to diagnose *T. vaginalis*, each with different diagnostic accuracies and costs. Fresh microscopic examination of vaginal content for direct parasite detection is a point-of-care (POC) diagnostic tool [17] that presents a wide range (35–80%) of sensitivity, depending on the experience of the evaluator [2]. Detection of *T. vaginalis* antigens is another POC method, with the latest-generation OSOM test (Genzyme Diagnostics, Cambridge, MA, USA) showing greater sensitivity than microscopy [18].

More commonly used methods include culture and nucleic acid amplification tests (NAATs), with higher sensitivity and specificity rates (92% and 98% for InPouch culture; 98.1% and 98.3% for NAATs—APTIMA, respectively) [2]. Regarding the culture methods, Diamond’s medium or the InPouch TV culture system (BioMed Diagnostics, White City, OR, USA) require incubation at 37 °C for 3 to 7 days before *T. vaginalis* can be observed microscopically. While culture techniques are highly sensitive and specific, some disadvantages should be considered; they are time-consuming, expensive, labor-intensive, and less sensitive than molecular biology methods like NAATs [19]. NAATs are high-sensitivity tests that can detect *T. vaginalis* DNA or RNA through the amplification of target sequences and detection of amplified material [2]. Samples used for diagnosing bacterial endocervicitis are also suitable for detecting *T. vaginalis* nucleic acids. The APTIMA TV kit (Gen-Probe, San Diego, CA, USA), which is FDA-approved, is one such example and can be performed either automatically or on a semi-automatic platform, though its high cost limits its availability in Brazil’s Unified Health System (SUS).

As an alternative technique, there is the Papanicolaou test, or Pap test (cytopathological examination), which is frequently used to diagnose trichomoniasis, despite its relatively low accuracy for this purpose (sensibility: 60–95%, specificity 98–100%) [2,5]. The use of this test to detect *T. vaginalis* infection can be explained by its widespread use and accessibility, as women undergo preventive cervical cancer screening periodically.

In Brazil, routine screening by Pap test is recommended for women aged 25 to 64 years who have already had sexual intercourse and should be conducted every three years after two consecutive negative annual exams [20,21]. Brazil aims to achieve 85% coverage of Pap smears in the recommended age group by 2030 [22]. The Pap smear is low-cost, safe, easy to perform, and generally well accepted by the female population. Additionally, it is available at all basic health units [20]. In our municipality, the prevalence of this STI among women undergoing Papanicolaou testing is unknown.

Considering the importance and relevance of STIs and the impacts of diagnosis and treatment on breaking the epidemiological chain, knowing the prevalence of *T. vaginalis* infection in the local population is crucial. Through cervical cytology analysis, this study aimed to determine the prevalence of *T. vaginalis* infection in women attending the cervical cancer precursor lesions screening program in Botucatu in São Paulo, Brazil.

## 2. Materials and Methods

This descriptive and retrospective study included 23,735 women who participated in the cervical cancer screening program conducted at primary health care units in Botucatu, São Paulo, during 2019 and 2022. These years were selected because they represent one year before the COVID-19 pandemic and another shortly after the peak of the most severe health crisis in recent years. All women met the following eligibility criteria: non-pregnant, had not received treatment for lower genital tract infections in the past 30 days, and had abstained from sex for at least 72 h prior to the exam.

Cytopathological exams were evaluated according to the Bethesda System [23]. Cervical smear samples were classified as satisfactory (with a representative number of cells that allow for a diagnostic conclusion), satisfactory with limitations (due to technical issues like staining or trauma during collection), or unsatisfactory. In this case, it was not possible to perform the evaluation, and the test was repeated. Smears were initially assessed at 40× magnification and subsequently at 100× magnification, performing the standard reading of all fields. In cases where greater precision was required to evaluate the cell or any abnormalities, 400× magnification was used [24].

Data regarding the total number of cervical cancer screening tests performed in the municipality during the years of interest were collected from the cancer information database (Siscan). The selected data were then compiled into a Microsoft Excel spreadsheet containing each patient’s name, age, medical record number, National Registry of Health Establishments (CNES number), characterization of their vaginal microbiota, presence of inflammation, and presence of *T. vaginalis* infection according to cervical cytology results and cytopathological diagnosis.

To characterize the sociodemographic and behavioral data of the patients, the following data were retrieved from the test requisition forms: age, marital status, education, occupation, family income, self-declared ethnicity, and use of contraceptives at the time of the cytological exam.

A descriptive analysis was conducted, and comparisons were performed between 2019 and 2022 using a chi-square test and Student’s *t*-test using the SigmaPlot version 13.0 software. The significance level adopted was 5%. The present study was approved by the Research Ethics Committee of the Faculty of Medicine of Botucatu—UNESP (CAAE nº 69754323.6.0000.5411). The application of the informed consent form (ICF) was waived due to the retrospective nature of the study.

## 3. Results

The overall prevalence of *T. vaginalis* infection was 0.72% (170/23,735) [0.84% (107/12,598) in 2019, and 0.57% (63/11,137) in 2022], *p* < 0.001. The mean age of patients with trichomoniasis was 42 years (±11.2), 75% (127/170) self-identified as white, 43% (73/170) reported being married or living in a stable union, 16% (27/170) used oral contraceptives, 40% (68/170) had not completed primary education, and 63% (107/170) were employed (Table 1).

Regarding the vaginal microbiota, only 15.3% (26/170) of the cytology exams with *T. vaginalis* infection showed a predominance of lactobacilli species. The remaining exams showed a predominance of supra cytoplasmic bacilli suggestive of *Gardnerella vaginalis* and *Mobiluncus sp.* or cocacceae microbiota. Inflammation was present in 82% (140/170) of the smears. Precursor lesions of cervical cancer were detected in 0.05% of patients with trichomoniasis, including ASC-US (atypical squamous cells of undetermined significance), ASC-H (atypical squamous cells that cannot exclude high-grade squamous intraepithelial lesion), LSILs (low-grade intraepithelial squamous lesions) and HSILs (high-grade intraepithelial squamous lesions) (Table 2).

Age, ethnicity, marital status, employment, place of birth, and gynecological data, including the use of oral contraceptives and the presence of inflammation in cervical cytology, were statistically similar between the years evaluated (Table 1). Infection with *T. vaginalis* was significantly associated with alterations in vaginal microbiota described in oncotic cytology, and the frequency of cytological alterations was significantly higher in 2019 compared to 2022, as displayed in Table 2.

## 4. Discussion

We aimed to determine the general prevalence of *T. vaginalis* in the female population of Botucatu in São Paulo, Brazil, through the analysis of cervical cytology from women attending the cervical cancer screening program during the years 2019 and 2022. The detected prevalence was 0.84% in 2019 and 0.57% in 2022, which was significantly lower shortly after the peak of the COVID-19 pandemic. Several factors could explain these findings, and the effects of social isolation imposed by the COVID-19 pandemic should not be overlooked. Using the highly sensitive Diamond’s medium culture, our group had previously reported a prevalence of 1.3% in a similar population, with a 4-fold increase among women with vaginal dysbiosis [25]. This difference is probably explained by the distinct sensitivities of the diagnostic methods used.

Studies conducted in the USA indicated an overall prevalence of *T. vaginalis* of 3.1% among non-white single patients aged 14 to 49, with lower educational levels, using PCR as a diagnostic tool [4]. Other studies in Brazil reported significantly higher prevalence rates, even when using cytology as a detection method. In Recife, Lima et al. detected a prevalence of 18.5% in women of reproductive age (20 to 30 years) who were married and had completed primary education [6]. Similarly, a study conducted in Maranhão detected a trichomoniasis rate of 19.0% in women aged 30 to 49 years old who were non-white, single, and had completed secondary education [26]. Such differences may be partially explained by environmental and social context. The rate herein reported is similar to that of a study conducted in the state of Pará, which also used cytology exams and detected a prevalence of 0.68% [10]. The sociodemographic data of our patients corroborate this low prevalence, as the mean age was above 25 years, and a significant portion of participants reported being married or living in a stable union and were employed.

Corroborating the literature, vaginal dysbiosis, which is an important factor for susceptibility to acquiring STIs, was present in 77% of women with trichomoniasis. Interestingly, the presence of the protozoon can profoundly alter the composition of the vaginal microbiota. According to Margarita et al., *T. vaginalis* infection establishes complex interactions with the microbial community of the vagina, many of which are incompletely understood [27]. Brotman et al. showed that dysbiotic microbiome (community state type CST-IV) is associated with 8-fold increased odds of detecting *T. vaginalis* compared with women presenting an *L. crispatus*-dominated microbiome (OR: 8.26, 95% CI: 1.07–372.65) [28]. Accordingly, in a previous study, we described the evidence for a 4-fold increased risk of *T. vaginalis* infection among women who had abnormal vaginal microbiota [25].

The elevated presence of the inflammatory infiltrate detected (82%) is a common feature of trichomoniasis once the establishment of *T. vaginalis* depends on the adherence of the trophozoite to the epithelial cells, provoking tissue damage, neutrophilic infiltration, and the production of inflammatory cytokines [29]. Another notable finding is the higher prevalence of abnormal cytological findings associated with *T. vaginalis* infection in 2019 compared to 2022.

A recent systematic review and meta-analysis demonstrated that 56.9% of *T. vaginalis* infections are asymptomatic among women in low- and middle-income countries [30]. The scarcity of symptoms is one of the causes of the underdiagnosis and undertreatment of the condition, which can lead to complications. Two recent meta-analyses have evaluated the magnitude of the risk of cervical cancer associated with *T. vaginalis* infection and showed that these women presented a significantly higher risk of developing cervical cancer [31,32]. Although the authors highlight that there are regional and ethnic influences on this association, the persistent co-infection of *T. vaginalis* and HPV should be understood as an important risk factor for cervical cancer, since chronic inflammation and host immune imbalance are hallmarks of carcinogenesis.

A limitation of our study is that the prevalence rate of trichomoniasis we reported was probably underestimated, as it was diagnosed based on cervical cytology results rather than culture or NAAT techniques. Nevertheless, its importance relies on the large dataset collected that allows a better understanding of the epidemiologic panorama of *T. vaginalis* infection in our population using a widely available method. Detecting this STI is important as it allows for treatment. Adequate treatment of both the diagnosed patient and their sexual partner(s) is paramount to break the transmission chain. Assessing trichomoniasis rates can be a tool to support health policies aimed at reducing infection rates and reproductive consequences.

## 5. Conclusions

The present study showed a low prevalence of *T. vaginalis* infection in low-risk women attending the cervical cancer precursor lesion screening program in the Municipality of Botucatu in São Paulo, Brazil. Nevertheless, *T. vaginalis* screening through cervical cytology can be of importance in the absence of other diagnostic methods, as it can help to break the transmission chain.

## Figures and Tables

**Table 1 microorganisms-12-02032-t001:** Sociodemographic and clinical data from patients with trichomoniasis (2019 and 2022).

Variables	All (n = 170)	2019	2022	*p*
		n = 107	n = 63	
Age (years)	42.4 ± 11.2	41.6 ± 12.1	42.4 ± 11.2	0.630
Ethnicity				
White	127 (75%)	80 (75%)	47 (75%)	0.148
Not white	41 (24%)	26 (24%)	15 (24%)	
Unknown	2 (1%)	1 (1%)	1 (1%)	
Marital status				
Married/stable union	73 (43%)	53 (49%) *a*	20 (32%) *b*	0.054
Single	66 (39%)	35 (33%) *c*	31 (49%) *d*	
Divorced/widowed	27 (16%)	16 (15%)	11 (17%)	
Unknown	4 (2%)	3 (3%)	1 (2%)	
Years of study				
< 11 years	97 (57%)	61 (57%)	36 (57%)	0.935
≥ 11 years	61 (36%)	39 (37%)	22 (35%)	
Unknown	12 (7%)	7 (6%)	5 (8%)	
Employed				
No	63 (37%)	38 (35%)	25 (40%)	0.543
Yes	107 (63%)	69 (65%)	38 (60%)	
Use of contraceptive				
No	141 (83%)	86 (80%)	55 (87%)	0.356
Yes	27 (16%)	19 (18%)	8 (13%)	
Unknown	2 (1%)	2 (2%)	0	
Dysbiosis				
No	26 (15%)	18 (17%)	8 (13%)	<0.05
Yes	131 (77%)	81 (76%)	50 (79%)	
Unknown	13 (8%)	8 (7%)	5 (8%)	
Inflammation				
No	3 (2%)	2 (2%)	1 (1%)	0.99
Yes	140 (82%)	91 (85%)	49 (78%)	
Unknown	27 (16%)	14 (13%)	13 (21%)	
Place of birth				
Botucatu region	78 (46%)	48 (45%)	30 (48%)	0.624
Other regions	83 (49%)	55 (51%)	28 (44%)	
Unknown	9 (5%)	4 (4%)	5 (8%)	

Specific *p*-values for marital status: a × b: *p* = 0.004, c × d: *p* = 0.049, compared by X^2^.

**Table 2 microorganisms-12-02032-t002:** Cytological alterations detected in patients with trichomoniasis (2019 and 2022).

Lesions	Total	2019	2022	*p*
ASC-US	8	8		<0.05
ASC-H	2	1	1	
HSIL	1	1		
LSIL	1	1		

ASC-US: Atypical squamous cells of undetermined significance; ASC-H: atypical squamous cells of undetermined significance that cannot exclude HSILs; LSIL: low-grade squamous intraepithelial lesion; HSIL: high-grade squamous intraepithelial lesion.

## Data Availability

The data presented in this study are available in SISCAN at https://acesso.saude.gov.br/login (accessed on 5 December 2023). These data were derived from the following resources available in the public domain: DataSUS.

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
