# Peer review of "Prevalence of Trichomonas vaginalis Infection in Women Screened for Precursor Lesions of Cervical Cancer in a Brazilian Population"

_microorganisms, 2024, doi:10.3390/microorganisms12102032_

Round 1

Reviewer 1 Report

Comments and Suggestions for Authors

1. "Table 2. Sociodemographic and...". Table 2 does not include sociodemographic or clinical (physical examination data, symptoms, medical history, etc.) data.

2. Line 143-145. The authors compare the data of their previous study (1.3%) with the current one. But they do not indicate that the sensitivity of the Diamond method is 80-85%, and the sensitivity of the diagnosis of trichomoniasis by cytopathological studies (Pap test) is 60-65%, which may explain the difference of 1.3% detection in the previous study and 0.72% in this study.

3. Table 1. It is worth checking with a statistician, but it seems to me that for complete accuracy, it would be worth adding a p-value for each row or category in the table (for example, separately for "Married/Stable Union", "Single" etc.) to show whether the differences are statistically significant for each specific group.

4. In the discussion section, it is worth adding more about asymptomatic/chronic carriage of Trichomonas vaginali. And also about the connection between asymptomatic carriage of Trichomonas vaginali and cervical cancer.

Author Response

Dear editors and reviewers, 

We appreciate the editors and the reviewers for the attention and time taken during the evaluation of our manuscript entitled “Prevalence of Trichomonas vaginalis infection in women screened for precursor lesions of cervical cancer in a Brazilian population.” submitted to the Microorganisms editorial board for publication. The constructive criticism given by our peers is welcome and was considered during this revision. We now resubmit a revised version according to the recommendations received and take this opportunity to answer the comments pointed out.

Reviewer 1

  1. "Table 2. Sociodemographic and...". Table 2 does not include sociodemographic or clinical (physical examination data, symptoms, medical history, etc.) data.

We unadvisedly replicated the title from Table 1. We have corrected the title, as Table 2 indicates the cytological alterations detected in patients with trichomoniasis.

  1. Line 143-145. The authors compare the data of their previous study (1.3%) with the current one. But they do not indicate that the sensitivity of the Diamond method is 80-85%, and the sensitivity of the diagnosis of trichomoniasis by cytopathological studies (Pap test) is 60-65%, which may explain the difference of 1.3% detection in the previous study and 0.72% in this study.

We thank the reviewer for the observation, we included this comment in the text.

  1. Table 1. It is worth checking with a statistician, but it seems to me that for complete accuracy, it would be worth adding a p-value for each row or category in the table (for example, separately for "Married/Stable Union", "Single" etc.) to show whether the differences are statistically significant for each specific group.

As requested, a p-value for each row of the non-dichotomous variable (marital status) was added (legend of table 1).

  1. In the discussion section, it is worth adding more about asymptomatic/chronic carriage of Trichomonas vaginalis. And also about the connection between asymptomatic carriage of Trichomonas vaginalis and cervical cancer.

As requested, we added to the discussion the following exert: A recent systematic review and meta-analysis demonstrated that 56.9% of T. vaginalis infections are asymptomatic among women in low- and middle-income countries [Fortas et al., 2024]. The scarcity of symptoms is one of the causes of the underdiagnosis and undertreatment of the condition, which can lead to complications. Two recent me-ta-analyses have evaluated the magnitude of the risk of cervical cancer associated with T. vaginalis infection and showed that these women presented a significantly higher risk of developing cervical cancer [Fazlollahpour-Naghibi et al., 2023, Yanga et al., 2018]. Although the authors highlight that there are re-gional and ethnic influences on this association, the persistent co-infection of T. vaginalis and HPV should be understood as an important risk factor for cervical cancer, since chronic inflammation and host immune imbalance are hallmarks of carcinogenesis.

References:

Fortas C, Delarocque-Astagneau E, Randremanana RV, Crucitti T, Huynh BT. Asymptomatic infections with Chlamydia trachomatis, Neisseria gonorrhoeae, and Trichomonas vaginalis among women in low- and middle-income countries: A systematic review and meta-analysis. PLOS Glob Public Health. 2024;4(5):e0003226. Published 2024 May 23. doi:10.1371/journal.pgph.0003226

Fazlollahpour-Naghibi A, Bagheri K, Almukhtar M, Taha SR, Zadeh MS, Moghadam KB, et al. (2023) Trichomonas vaginalis infection and risk of cervical neoplasia: A systematic review and meta-analysis. PLoS ONE 18(7): e0288443. https:// doi.org/10.1371/journal.pone.0288443

Yang S, Zhao W, Wang H, Wang Y, Li J, Wu X. Trichomonas vaginalis infection-associated risk of cervical cancer: A meta-analysis. Eur J Obstet Gynecol Reprod Biol. 2018;228:166-173. doi:10.1016/j.ejogrb.2018.06.031

Reviewer 2 Report

Comments and Suggestions for Authors

The manuscript “ Prevalence of Trichomonas vaginalis infection in women screened for precursor lesions of cervical cancer in a Brazilian population” aimed to describe the Trichomonas vaginalis status in women attending a cancer screening program in Brazil. The authors included some relevant variables like disbioses, use of contraceptive and cytology. However, Results and Discussion sections are very limited. The authors might include further considerations in the manuscript.

Comments:

1) The abstract is missing an introduction.

2) Introduction section: please briefly report data about the performances of diagnostic procedures. In addition, many studies consider NAATs as the gold standard for trichomoniasis diagnosis, please revise the manuscript accordingly.

3) Materials and Methods: what's the target population of the mentioned cervical cancer screening program?

4) Results, "Regarding the vaginal microbiota, only 15.3% (26/170) of the cytology exams with T. vaginalis infection showed a predominance of lactobacilli species and inflammation was present in 82% (140/170) of the smears": and the remaining part?

5) A further evaluation of the variables and lesion status between individuals with and without trichomoniasis should be considered. Discuss the obtained results accordingly.

6) Is HPV status available for the enrolled individuals?

7) Regarding the transmission chain, please cite the potential role of males in T. vaginalis transmission.

8) Check the manuscript for typos.

Comments on the Quality of English Language

Minor editing of English language required.

Author Response

We appreciate the editors and the reviewers for the attention and time taken during the evaluation of our manuscript entitled “Prevalence of Trichomonas vaginalis infection in women screened for precursor lesions of cervical cancer in a Brazilian population.” submitted to the Microorganisms editorial board for publication. The constructive criticism given by our peers is welcome and was considered during this revision. We now resubmit a revised version according to the recommendations received and take this opportunity to answer the comments pointed out.

Reviewer 2

Comments:

1) The abstract is missing an introduction.

We appreciate the observation. We added an introduction to the abstract.

2) Introduction section: please briefly report data about the performances of diagnostic procedures. In addition, many studies consider NAATs as the gold standard for trichomoniasis diagnosis, please revise the manuscript accordingly.

We included data about the sensitivity of the diagnostic methods, as requested.

3) Materials and Methods: what's the target population of the mentioned cervical cancer screening program?

In Brazil, routine screening by Pap smear (cytopathological examination) is recommended for women aged 25 to 64 years who have already had sexual intercourse and should be conducted every three years after two consecutive negative annual exams (INCA, 2016, 2021). Brazil aims to achieve 85% coverage of Pap smears in the recommended age group by 2030 (Ministério da Saude, 2021). The Pap smear is low-cost, safe, and easy to perform and generally well accepted by the female population. Additionally, it is available at all Basic Health Units (INCA 2016). We included this information in the manuscript.

4) Results, "Regarding the vaginal microbiota, only 15.3% (26/170) of the cytology exams with T. vaginalis infection showed a predominance of lactobacilli species and inflammation was present in 82% (140/170) of the smears": and the remaining part?

The remaining exams showed a predominance of supra cytoplasmic bacilli suggestive of Gardnerella vaginalis and Mobiluncus sp. or cocacceae microbiota. This was included in the Results section, as well as a discussion on the matter.

5) A further evaluation of the variables and lesion status between individuals with and without trichomoniasis should be considered. Discuss the obtained results accordingly.

The main goal of the study was to determine the prevalence of T. vaginalis in our female population using a widely available method and to characterize this population, including the prevalence of precursor lesions. Thus, evaluating lesions status of the remaining 23,565 patients (without trichomoniasis), although interesting, is out of our scope.

6) Is HPV status available for the enrolled individuals?

The status of HPV infection and its genotyping is not available in the cervical cancer screening program in our country as a government program. Due to this reason, we do not have this information to incorporate into the manuscript.

7) Regarding the transmission chain, please cite the potential role of males in T. vaginalis transmission.

We included the following sentence: Adequate treatment of both the diagnosed patient and their sexual partner(s) is paramount to break the transmission chain.

8) Check the manuscript for typos.

The manuscript was checked for typos.

References

BRASIL. Instituto Nacional de Câncer José Alencar Gomes da Silva.Coordenação de Prevenção e Vigilância. Divisão de Detecção Precoce eApoio à Organização de Rede. Diretrizes brasileiras para o rastreamento do câncer do colo do útero.  2. ed. rev. atual. – Rio de Janeiro: INCA, 2016. Disponível em: https://www.inca.gov.br/sites/ufu.sti.inca.local/files/media/document/diretrizesparaorastreamentodocancerdocolodoutero_2016_corrigido.pdf

INSTITUTO NACIONAL DE CÂNCER JOSÉ ALENCAR GOMES DA SILVA (INCA). Detecção precoce do câncer. – Rio de Janeiro: INCA, 2021. Disponível em: https://www.inca.gov.br/sites/ufu.sti.inca.local/files/media/document/deteccao-precoce-do-cancer.pdf

BRASIL. MINISTÉRIO DA SAÚDE PLANO DE AÇÕES ESTRATÉGICAS PARA O ENFRENTAMENTO DAS DOENÇAS CRÔNICAS E AGRAVOS NÃO TRANSMISSÍVEIS NO BRASIL 2021-2030. Ministério da Saude; 2021. Disponivel em https://www.gov.br/saude/pt-br/centrais-de-conteudo/publicacoes/svsa/doencas-cronicas-nao-transmissiveis-dcnt/09-plano-de-dant-2022_2030.pdf

Brotman RM, Bradford LL, Conrad M, et al. Association between Trichomonas vaginalis and vaginal bacterial community composition among reproductive-age women. Sex Transm Dis. 2012;39(10):807-812. doi:10.1097/OLQ.0b013e3182631c79

Margarita V, Fiori PL and Rappelli P (2020) Impact of Symbiosis Between Trichomonas vaginalis and Mycoplasma hominis on Vaginal Dysbiosis: A Mini Review. Front. Cell. Infect. Microbiol. 10:179. doi: 10.3389/fcimb.2020.00179

Ibáñez-Escribano A, Nogal-Ruiz JJ. The Past, Present, and Future in the Diagnosis of a Neglected Sexually Transmitted Infection: Trichomoniasis. Pathogens. 2024;13(2):126. Published 2024 Jan 29. doi:10.3390/pathogens13020126

Reviewer 3 Report

Comments and Suggestions for Authors

Dear authors 

I was happy to review your manuscript "Prevalence of Trichomonas vaginalis infection in women  screened for precursor lesions of cervical cancer in a Brazilian population."

The authors determined the prevalence of Trichomonas vaginalis infection in women screened for cervical cancer precursor lesions from Brazil. There was an immpresive number of 23,735 women who attended the cervical cancer screening, for several years. Prevalence of T. vaginalis infection was 0.84% in 2019 and 0.57% in 2022.  Cytological analysis revealed precursor lesions of cervical cancer in 0.05% of patients with trichomoniasis, including  ASC, LSIL and HSIL. In  low-risk women screened for precursor lesions of cervical cancer there was  a low prevalence of infection by T. vaginalis.

As suggestions I recommend to discuss more in the discussion section about the possible link of T vaginalis on Cervical cancers.

I believe the introduction section, material and methods are fine.

The result are clearly presented .

The conclusion support the results,

Author Response

We appreciate the editors and the reviewers for the attention and time taken during the evaluation of our manuscript entitled “Prevalence of Trichomonas vaginalis infection in women screened for precursor lesions of cervical cancer in a Brazilian population.” submitted to the Microorganisms editorial board for publication. The constructive criticism given by our peers is welcome and was considered during this revision. We now resubmit a revised version according to the recommendations received and take this opportunity to answer the comments pointed out.

Reviewer 3

As suggestions I recommend to discuss more in the discussion section about the possible link of T. vaginalis on Cervical cancers.

We thank the reviewer for the valuable consideration. We added the following discussion: As requested, we added to the discussion the following exert: A recent systematic review and meta-analysis demonstrated that 56.9% of T. vaginalis infections are asymptomatic among women in low- and middle-income countries [Fortas et al., 2024]. The scarcity of symptoms is one of the causes of the underdiagnosis and undertreatment of the condition, which can lead to complications. Two recent me-ta-analyses have evaluated the magnitude of the risk of cervical cancer associated with T. vaginalis infection and showed that these women presented a significantly higher risk of developing cervical cancer [Fazlollahpour-Naghibi et al., 2023, Yanga et al., 2018]. Although the authors highlight that there are re-gional and ethnic influences on this association, the persistent co-infection of T. vaginalis and HPV should be understood as an important risk factor for cervical cancer, since chronic inflammation and host immune imbalance are hallmarks of carcinogenesis.

References:

Fortas C, Delarocque-Astagneau E, Randremanana RV, Crucitti T, Huynh BT. Asymptomatic infections with Chlamydia trachomatis, Neisseria gonorrhoeae, and Trichomonas vaginalis among women in low- and middle-income countries: A systematic review and meta-analysis. PLOS Glob Public Health. 2024;4(5):e0003226. Published 2024 May 23. doi:10.1371/journal.pgph.0003226

Fazlollahpour-Naghibi A, Bagheri K, Almukhtar M, Taha SR, Zadeh MS, Moghadam KB, et al. (2023) Trichomonas vaginalis infection and risk of cervical neoplasia: A systematic review and meta-analysis. PLoS ONE 18(7): e0288443. https:// doi.org/10.1371/journal.pone.0288443

Yang S, Zhao W, Wang H, Wang Y, Li J, Wu X. Trichomonas vaginalis infection-associated risk of cervical cancer: A meta-analysis. Eur J Obstet Gynecol Reprod Biol. 2018;228:166-173. doi:10.1016/j.ejogrb.2018.06.031

Round 2

Reviewer 2 Report

Comments and Suggestions for Authors

Concerns addressed.